# Coupling between Sequence-Mediated Nucleosome Organization and Genome Evolution

**DOI:** 10.3390/genes12060851

**Published:** 2021-06-01

**Authors:** Jérémy Barbier, Cédric Vaillant, Jean-Nicolas Volff, Frédéric G. Brunet, Benjamin Audit

**Affiliations:** 1Institut de Génomique Fonctionnelle de Lyon, Univ Lyon, CNRS UMR 5242, Ecole Normale Supérieure de Lyon, Univ Claude Bernard Lyon 1, F-69364 Lyon, France; jeremy.barbier@ens-lyon.fr (J.B.); frederic.brunet@ens-lyon.fr (F.G.B.); 2Laboratoire de Physique, Univ Lyon, ENS de Lyon, CNRS, F-69342 Lyon, France; cedric.vaillant@ens-lyon.fr

**Keywords:** DNA sequence-encoded nucleosome ordering, nucleosome depleted regions, DNA sequence mutation, chromatin evolution

## Abstract

The nucleosome is a major modulator of DNA accessibility to other cellular factors. Nucleosome positioning has a critical importance in regulating cell processes such as transcription, replication, recombination or DNA repair. The DNA sequence has an influence on the position of nucleosomes on genomes, although other factors are also implicated, such as ATP-dependent remodelers or competition of the nucleosome with DNA binding proteins. Different sequence motifs can promote or inhibit the nucleosome formation, thus influencing the accessibility to the DNA. Sequence-encoded nucleosome positioning having functional consequences on cell processes can then be selected or counter-selected during evolution. We review the interplay between sequence evolution and nucleosome positioning evolution. We first focus on the different ways to encode nucleosome positions in the DNA sequence, and to which extent these mechanisms are responsible of genome-wide nucleosome positioning in vivo. Then, we discuss the findings about selection of sequences for their nucleosomal properties. Finally, we illustrate how the nucleosome can directly influence sequence evolution through its interactions with DNA damage and repair mechanisms. This review aims to provide an overview of the mutual influence of sequence evolution and nucleosome positioning evolution, possibly leading to complex evolutionary dynamics.

## 1. Introduction

To fit in the nucleus of each cell, eukaryotic DNA needs to be highly compacted. This compaction is achieved by the formation of a protein-DNA complex called chromatin [1]. The first level of compaction consists of the wrapping of ∼146 bp of DNA around an octamer of four core histone proteins (H2A, H2B, H3 and H4), forming a nucleosome [2]. In the nucleosome, the DNA is wrapped almost twice around the core histone octamer (a tetramer of (H3-H4)${}_{2}$ flanked by two dimers of H2A-H2B), with contact points between DNA and the histone proteins every ∼10 bp [3,4]. The mid-point of the complexed DNA is called the dyad, and serves as a reference to specify nucleosome positions. The nucleosome repeat length (NRL), that represents the distance between two consecutive nucleosome dyads, ranges from 155 bp in fission yeast [5] to about 240 bp in echinoderm sperm [6]. Taking into account the length of DNA wrapped in each nucleosomes, there is thus a high density of nucleosome in living cells regardless of the cell type or organism, with at least two third of the genome participating in a nucleosome. Nucleosomes come in several forms. Core histones may carry post-translational modifications (PTMs), such as methylation, acetylation or phosphorylation occurring mostly in the N-terminal tail of histones (e.g., tri-methylation of histone H3 lysine 9, also known as H3K9me3). Histone cores may also contain histone variants, which are alternative histone proteins encoded by genes that appeared throughout the evolution of Eukaryotes [4,7,8]. PTMs and histone variants are associated with different chromatin states of genome compaction and genome regulation and have thus received most of the attention in chromatin biology studies. Nevertheless, the precise position of nucleosomes on the DNA is also of great importance [1]. Indeed, the accessibility of DNA to non-histone chromatin factors like transcription and replication factors is modulated by nucleosome occupancy, with nucleosomal DNA being considerably less accessible to theses factors than the naked “linker” DNA between nucleosomes. From a collective perspective, the position of nucleosomes relative to each other is also associated to chromatin state, probably in relation to higher order chromatin compaction. Indeed, actively transcribed genomes where chromatin needs to be open and accessible tend to have shorter NRL (ranging from 160 to 189 bp in yeast, embryonic stem cells and tumour cells for example) than transcriptionaly inactive genomes (NRL ranging from 190 to 240 in chicken erythrocytes and echinoderm sperm for example) [9]. This distinction has also been made within the human genome, where the NRL of active genes is way shorter (178 bp) than the NRL of repressed or heterochromatic non-coding sequences (206 bp) [10]. However, there are exceptions to this rule. For example, in higher eukaryotes, telomeric DNA is packaged in nucleosomes with a NRL 20–40 bp shorter than the NRL of bulk nucleosome [11]. This has been observed in vertebrates [12,13,14,15,16] but also in sea urchin [16], and several plant species [17,18,19]. The position of nucleosomes on the DNA and relative to each other is thus crucial for genetic functions, because it modulates the efficiency of trans-acting factors such as the transcription machinery [1,2,20]. Nucleosomal positioning on DNA depends on various factors, including DNA sequence effects, competition for DNA such as with transcription factors, and remodeling by ATP-dependent enzyme [21]. Notably, the DNA sequence has an important contribution to nucleosomal positioning at the genome scale [1,10,21,22]. Nucleosome positions are thus to some significant extend a sequence-encoded feature that have a functional role in genomes (as modulator of the accessibility to DNA). As other sequence-encoded functional features (such as genes), nucleosome positions can then be selected during evolution. In other words, sequences could be selected not for their direct coding properties as genes, but for their abilities to favor or impair nucleosome formation at specific loci, directly impacting their accessibility to external regulatory factors. Selection of sequences for their nucleosomal affinity has been described in several species such as yeasts [23,24] but also in more complex organisms like maize [25,26] or human [27,28,29]. Note that the repositioning of nucleosomes according to the evolution of sequences can also occur in a neutral scenario leading to possible drifts of nucleosome positions [30]. Interestingly, the nucleosome itself also shapes the evolution of sequences by interacting with DNA damage and repair mechanisms, leading to biased mutational patterns inside and around nucleosomes [31]. Here, we will review some of the findings about these mechanisms, focusing first on how nucleosome positions are encoded in the DNA sequence, then on how sequence nucleosomal properties have been selected during evolution, and finally, on how the nucleosome directly modulates mutational patterns. This provides an opportunity to discuss the mutual feedback between the evolution of DNA sequence and chromatin organization at a genomic scale.

## 2. How Is Nucleosome Positioning Encoded in the DNA Sequence?

### 2.1. DNA Sequence Does Influence Nucleosome Positioning

Using SELEX (Systematic Evolution of Ligands by EXponential enrichment) experiments on synthetic and genomic DNA with the core histone proteins as ligands, it was shown that the DNA sequence does influence the affinity of a DNA fragment for histones up to a 5000-fold range [32,33,34,35]. In such experiments, an excess of DNA fragments of variable sequence compete for a ligand. The DNA-ligand complexes are then extracted, DNA fragments are purified, amplified and brought back into competition with the same ligand, a process repeated several times to purify sequences with the highest affinities for the ligand of interest. Lowary and Widom used this approach with synthetic DNA fragments and core histone proteins as ligands to select from a random set of sequences the ones with the highest affinities for the nucleosome [33]. It revealed the existence of sequences with unexpectedly high affinity for the histone octamer. Similar experiments were also performed with fragments extracted from genomic DNA. It showed that their affinity for histones had a much narrower range than random DNA fragments [32,35]. These experiments clearly indicate that the DNA sequence matters on how easily a nucleosome can be formed and so where nucleosome are intrinsically positioned along chromosomes. The sequence-encoded nucleosome positioning can therefore be seen as a basal “ground state” that can be “remodeled” in vivo by the site-specific recruitment and (energy consuming) action of trans-acting factors to establish at proper times and positions an “epigenetic” reversible nucleosome positioning pattern, either permissive or repressive for genome activity. As demonstrated by Parmar et al. [36] when considering a composite model of nucleosome positioning that accounts for both sequence effects and ATP-dependent remodelers and as evidenced by experiments [37], sequence effects are indeed sufficiently strong to control the first steps of the relaxation dynamics of the nucleosomal array after strong perturbation, i.e., in a transient phase of non or weak activity of remodelers. Strikingly, nucleosomal pattern in germ cells where remodelers activity is reduced has been shown to be mostly controlled by the DNA sequence [38]. in vitro nucleosome reconstitution experiments on the yeast genome further demonstrated that ATP was required to obtain a nucleosome positioning pattern that deviate from the sequence encoded pattern and resemble the native pattern [39]. All these results suggest that the primary sequence is a parameter that needs to be taken into account in nucleosome positioning studies, even if sequence effects can be refined or even overridden in vivo by other factors such as ATP-dependent remodelers.

Technical progresses made it possible to decipher DNA sequence-mediated effects genome-wide, mainly with experiments such as MNase-seq, in which the chromatin is digested with an enzyme (the micrococcal nuclease, MNase) that cuts and digests the naked linker DNA between nucleosomes [40,41,42]. After histone removal, the remaining DNA can be sequenced with high-throughput sequencing techniques, and the alignment of the reads on the reference genome provides information about the genome-wide positioning of nucleosomes [10,42,43,44,45,46,47]. Such genome-wide mapping of nucleosomes has been established in vivo in various species, including yeast [43,47,48], human [10,44,45], fly [49], plants [25,50], mouse [51], and the nematode *Caernorhabditis elegans* [52], but also in vitro [10,53]. The availability of such experimental data has been reviewed by Teif [54]. Comparison of in vivo and in vitro nucleosome maps revealed a high consistency between in vitro and in vivo genome-wide positioning of nucleosomes [10,53,55]. These results showed that the sequence effects are relevant even in vivo in the presence of external factors influencing nucleosomal positioning. Indeed, the sequence-directed nucleosome positioning is directly observed from in vitro data, because chromatin is reconstituted from DNA and histones only, without any other external factors such as remodelers found in vivo. Accordingly, models established from in vitro genome-wide reconstitution of chromatin predict rather well in vivo nucleosome positioning [22,53,55,56,57,58,59,60], corroborating the hypothesis that the DNA sequence plays a major role among the different factors influencing the position of nucleosomes [61]. During the past 40 years, attempts to describe the sequence-directed nucleosomal positioning showed that one needs to consider two types of mechanisms (Figure 1): (i) positioning mechanisms where DNA motifs at specific location accommodate DNA wrapping in the nucleosome, for example by favoring certain dinucleotides at contact points between DNA and histones; and (ii) inhibiting mechanisms, with sequences such as poly(dA:dT) preventing nucleosome formation [1].

### 2.2. Sequence Motifs with 10 Base Pair Periodicity as Nucleosome
Positioning Signals

In the 1980s, the analysis of 32 coding and non-coding sequences (representing about 36,000 nucleotides) that were known to fold in chromatin-like structures (i.e., nucleosomes) exhibited a periodicity of ∼10.5 base pair (bp) in the distribution of dinucleotides along their sequences [62]. Dinucleotides GG, TA, TG and TT were found to be the strongest contributors to this observed periodicity. In other words, in sequences that fold in chromatin-like structures, dinucleotides GG, TA, TG and TT tend to be regularly spaced by 10 or 11 bp whereas other dinucleotides are more randomly positioned. Interestingly, no 10.5 bp periodicity was found for prokaryotic sequences. Further analysis showed a symmetry in the phasing of the preferential positionning of complementary dinucleotides within the 10.5 bp periodicity [63]. An explanation proposed for these observations was about the affinity of the DNA sequence for histone core. It was suggested that sequence periodicity and their symmetries facilitates the bending of the DNA molecule around the nucleosome core histones proteins [62,63]. It was even expected that it would be possible to predict nucleosome positioning from these sequence properties.

The “periodicity model” successfully predicted the curved shape of a 423 bp DNA restriction fragment containing a strong periodicity of AA and TT dinucleotides [64]. Sequence-encoded bending of DNA was explored in several studies [65,66,67,68], from which nucleosomal DNA bending tables were derived. In the nucleosome, A/T-rich sequences are preferred where the minor groove is facing inward, and G/C-rich sequences where it is facing outward of the structure [67]. In addition, homopolymers tend to be excluded from the nucleosome, especially from the dyad position [65,66,67,68]. Finally, it was observed that linker DNA regions between nucleosomes are cut poorly by DNAse I enzyme, that is known to cut poorly in homopolymers, probably revealing their strong occurrence in linker DNA [68], in accordance with the previous observation.

The sequence periodicities described here facilitate the bending of DNA around the histone octamer to form a nucleosome. Such sequences could have a positioning effect. During the course of evolution, some selective pressure could have acted on genomes to select those sequences at specific loci where the presence of a nucleosome is necessary. Periodicities associated to nucleosomal sequences have been found in several species, in chicken, but also in yeast, human and worm [53,56,61,68,69,70]. However, among genomic sequences, even the most powerful positioning sequences only have a weak positioning power [33]. Sequences optimized for wrapping into the nucleosome, like the sequence of the clone 601 established by Lowary and Widom in their SELEX experiment on artificial DNA [33], are not found in genomic DNA. In addition, the global positioning power of genomic DNA is not much higher than that of random DNA sequences [33]. Thus, positioning sequences and their periodicities in the dinucleotide distributions fail to explain the genome-wide sequence-encoded nucleosomal positioning [33]. However, periodic distribution of sequence motifs is not the only way to encode nucleosome position.

### 2.3. Sequence-Encoded Nucleosome Depleted Regions and Statistical Positioning

In yeast, it has been showed that promoters are enriched in what are called nucleosome-depleted regions (NDRs) [43]. In several yeast species, these NDRs are found both in vivo and in vitro, indicating that they are directly encoded in the DNA sequence, mainly through poly(dA:dT) sequences that are known to inhibit nucleosome formation [1]. The strength of the depletion depends mainly on the length and purity of the poly(dA:dT) sequence [1], allowing a fine tune regulation of gene expression in yeast [71]. Positioning of nucleosomes can arise from these NDRs, following a statistical positioning model [72,73], where nucleosomes stack against a fixed object (either a NDR or a highly positioned nucleosome) that serves as an anchor, forming an array of positioned nucleosomes (Figure 1). The closer a nucleosome is to the anchor, the better it is positioned. Thus, counter-intuitively, sequence-encoded nucleosome positioning could arise not from positioning sequences but rather from anti-positioning sequences that anchor the position of nucleosomal arrays. In the case of yeast promoters, if NDRs are observed both in vivo and in vitro, arrays of nucleosomes are only observed in vivo, on the side of the transcribed units [74]. In this case, the in vivo nucleosomal organization results from the combination of the sequence effect (mainly specifying the NDRs and probably the +1 nucleosomes) and the ATP-dependent chromatin remodelers (for the ordering of nucleosomes). Another type of arrays of nucleosomes relying only on sequences have been observed in yeast, where nucleosomes are confined between sequence-encoded NDRs when these NDRs are close to one another [55,57]. Indeed, when two NDRs are close enough to each other, constraints appear on the nucleosomal positioning, mainly because of the exclusion interaction between nucleosomes since two nucleosomes cannot superimpose. For example, if two sequence-encoded NDRs are separated by a distance of about 300 bp (∼2 nucleosomes), and one nucleosome is formed between the NDRs, it can be formed quite anywhere along the 300 bp. However, if 2 nucleosomes are formed, taking about 147 bp each, then the possibilities are greatly reduced and preferential positioning appears. Sequence-encoded arrays of nucleosomes can thus result from sequence-encoded NDRs and a high density of nucleosomes. This “statistical positioning between NDRs” model was experimentally validated with atomic force microscopy (AFM) visualization of nucleosome positioning along a DNA fragment bounded by two sequence-encoded NDRs separated by a two-nucleosomes long distance [55,75]. When either one or two nucleosomes were reconstituted on this fragment, single nucleosomes were observed anywhere between the barriers, but as predicted, the position of nucleosome pairs were very constrained.

In human, part of the genome-wide nucleosomal positioning follows this scenario of statistical positioning between NDRs [28,76]. Indeed, a physical model of nucleosome formation based on sequence-dependent bending properties of the DNA double helix revealed about 1.6 million nucleosome-inhibiting energy barriers (NIEBs) along the human genome. These NIEBs correspond to NDRs, both among in vivo and in vitro data. In both conditions, when NIEBs are close enough to each other (about four nucleosomes or less), a constrained positioning of nucleosomes is observed, just as described above in yeast. The in vitro observation indicates that this positioning is not dependent of the action of remodelers, but relies only on the sequence-encoded NIEBs/NDRs and high density of nucleosomes. in vitro map of nucleosomes also showed that a nucleosome-favoring sequence flanked by two nucleosome-deterring sequences can form what is called a “container” site in which a nucleosome is trapped [10]. Taken alone, each of these sequences do not have any significant positioning or anti-positioning power, but taken together, they form a highly positioned nucleosome at a specific locus. These container sites were also found in the in vivo nucleosomes maps, where they can serve as anchors to form nucleosomal arrays by stacking of the other nucleosomes against the well positioned one. The situation is similarly found at the promoters of yeast genome: a fixed object (here, a highly positioned nucleosome, a NDR in yeast) serves as an anchor for regularly spaced nucleosomal arrays. The difference is that the formation of the array is not associated with transcription as in yeast. However, these arrays are also only observed in vivo, indicating that if the anchor is sequence-encoded, the action of remodelers is needed to fluidify the movement of nucleosomes and allow statistical positioning. Note that isolated NIEBs can also serve as anchors: two to three positioned nucleosomes have been observed on their borders in human, both in vivo and in vitro [28,76], illustrating that the “stacking against an anchor” model does not always need the activity of remodelers.

### 2.4. Predicting Nucleosomal Positioning from Sequences

Nucleosome occupancy encoded in the sequence can presumably be predicted through sequence-based modeling. This was achieved using mainly two types of approaches: bioinformatic models relying on machine learning [22,53,56,58], and physical models relying on energy calculations [55,57,59,60,77]. The general idea of the bioinformatic models is to detect, genome-wide, the sequence features associated with nucleosomal positioning. For example, the model detailed in [53] is based on an in vitro map of yeast nucleosomes. From this map, the sequence preferences for nucleosomes are extracted to establish a probabilistic model that assigns a score to each 147 bp fragment. This score is based on the 5-mers observed along the sequence of the fragment. From the score landscape, and taking into account the impossibility to superimpose two nucleosomes, nucleosomal positioning can be predicted. This approach reproduced well experimental mapping of nucleosomes [53]. A simpler approach has been developed in [22], in which the over 2000 parameters of [53] are reduced down to only 14 parameters. It was even claimed that a model taking into account only the GC content and poly(dA:dT) sequences is sufficient to achieve good predictions of nucleosome occupancy [22]. The GC content is tightly correlated to nucleosome occupancy [27,28]. It was in fact argued that the observation that the genomic GC content of Eukarya is way less variable than that of Bacteria and Archaea corroborates this observation. It was linked to the high level of conservation of histones between organisms, whereas nucleoid-associated proteins are more variable, possibly allowing wider range for genomic GC content between species [78]. The physical modeling approach was considered independently by different groups [55,57,60,79]. It is based on intrinsic bending properties of the DNA and thus, its ability to be wrapped around histone octamers. The idea is to compute the energy needed to deform all 147 bp DNA fragments from their intrinsic conformation to the helical conformation adopted in the nucleosome, based on tabulated sequence-dependent elastic parameters. This provides an energy landscape for the formation potential of nucleosomes along the genome. The dynamic assembly of histone octamers along the DNA chain is then modeled as a fluid of rods of finite extension (the DNA wrapping length around the octamer), binding and moving in the nucleosome formation potential and respecting the exclusion relationship between nucleosomes. The nucleosome occupancy profile can then be deduced given a temperature and a chemical potential allowing to fix the average nucleosome density to the experimentally determined value. Nucleosome occupancy based on our implementation of the model [55,57] fits well the experimental occupancy data in yeasts, in the nematode *C. elegans* and the fly *D. melanogaster* [55,59,80], and in human [28,76].

## 3. Nucleosome Positioning during Evolution

### 3.1. Nucleosome Position as a Darwinian Feature

Nucleosome occupancy influences the binding of transcription factors by controlling the accessibility to DNA [25]. The modulation of nucleosome occupancy is thus a critical feature for gene transcription regulation. Indeed, the distribution of nucleosomes around genes was associated with transcription levels in several species, including yeast [81], human [10,44], mouse [51], drosophila [49], and plants such as the thale cress [50], rice [50] and maize [26]. For example, highly expressed genes are associated with a more pronounced nucleosome depletion at their promoter than lowly expressed genes. The transcriptional changes during cell life processes such as differentiation, reprogramming, stress or even aging are associated with changes in nucleosome occupancy [82,83,84,85]. Modifying the nucleosome organization at some loci is thus expected to have either a positive or a negative impact on the fitness of an individual [86]. As nucleosome positions are at least partially sequence-encoded (Section 2), this strongly suggests that natural selection on DNA sequence could have an impact on the nucleosomal positioning. In other words, mutations could be selected or counter-selected, not for their direct effect on coding sequences, but for their influence on the position of nucleosomes at some specific loci, indirectly influencing features under selection such as gene expression. Following this hypothesis, natural selection could favor nucleosome inhibiting sequences where sequences need to be constantly available to transcription factors (at the regulating sequences of constitutive genes for example). It could also favor certain nucleosomal organization on the body of genes according to the basal level of transcription needed. The latter possibility question the compatibility between the nucleosomal and the genetic codes, to allow encoding of both a protein sequence and the nucleosomal organization in the same sequences. This compatibility has been explored by Eslami-Mossallam et al. [87], revealing the possibility of multiplexing genetic and mechanical information along a single sequence. Indeed, it is achievable to change the nucleosomal organization on the body of a gene without changing the protein(s) associated with the gene, thanks to the redundancy of the genetic code [87].

### 3.2. Nucleosome Positioning and the Evolution of Gene Regulation

In yeast, “growth genes” are identified as genes almost constantly expressed during growth, often associated with the metabolic pathways used in *ideal* growth conditions. In contrast, “stress genes” are genes expressed only in certain specific conditions, for example to respond to an environmental change. At the nucleosomal level, differences have been observed between growth and stress genes. The prediction of the nucleosomal organization at the promoter of these different types of genes in two yeast species, *Candida albicans* and *Saccharomyces cerevisiae*, showed that on average growth genes exhibit an intrinsically open chromatin at their promoter, when stress genes harbor a more closed patterns [23]. The experimental confirmation of the predicted organizations, both in vitro and in vivo, demonstrated that they are encoded directly in both genomes. Thus, in these two yeasts, we have two distinct sequence-encoded nucleosomal patterns associated with the two modes of gene expression. These two species display major metabolism differences when grown in a high glucose environment: *C. albicans* that grows mainly using respirative metabolism is identified as an aerobic yeast, as oppose to *S. cerevisiae* that grows mainly using fermentative metabolism, identified as an anaerobic yeast. From an evolutionary standpoint, orthologous genes associated with respiration are growth genes in the former, that switched to stress genes in the latter during the evolution of yeasts. By comparing the nucleosomal organization at the promoter of these genes in these two species, it was shown that they exhibit an intrinsically open chromatin in *C. albicans*, and a closed chromatin in *S. cerevisiae* [23]. This pattern was also observed in 10 other yeast species for which the nucleosome occupancy was predicted genome-wide from the DNA sequence. These results were confirmed experimentally with the direct comparison of experimental nucleosome positioning and gene expression data in the same 10 yeast species [24]. It showed that gain or loss of poly(dA:dT) tracts are associated with modifications of the nucleosomal organization at several phylogenetic branch points [24]. For example, the promoters of mitochondrial ribosomal protein (mRP) genes have lost their poly-A-like sequences in anaerobic yeasts, changing the chromatin organization on these genes from an open conformation (in aerobic yeasts) to a closed one (in anaerobic yeasts) [23,24]. These experiments show that in the course of yeast evolution, nucleosomes located at the promoter of genes have been repositioned, notably through the modification of the DNA sequence, and it was associated to a major change in yeast metabolisms, such as the switch from an aerobic to an anaerobic metabolism. This is a very good example of sequence selection not acting directly on coding properties, but for their affinity to nucleosomes, allowing a fine tuning of gene regulation from growth expression to stress expression pattern.

A similar dichotomy is present in multi-cellular organisms, such as maize, in the form of constitutive genes that are expressed regardless of the cell type, versus tissue-specific genes that are expressed only in some specific cell types. Sequences selected for nucleosomal positioning have been observed in this species [25,26]. In maize, the expression level between tissues show only minor differences in constitutive genes which contrast with tissue-specific genes that show higher differences. This difference shows that tissue-specific genes have higher transcriptional plasticity than constitutive genes. It was proposed that the sequence-encoded nucleosomal organization of each gene controls its transcriptional plasticity instead of directly its level of expression [25,26]. Indeed, the level of expression can change between cell types and conditions, particularly for tissue-specific genes. If the level of expression was directly sequence-encoded through nucleosomal positioning, transcriptional plasticity could not be achieved, since the gene sequence is the same in each cell and condition. In maize, the prediction from sequences of the nucleosomal organization of different set of genes showed that constitutive genes have the lowest sequence-encoded global nucleosome occupancy, while tissue-specific genes have the highest [26]. Compared to tissue-specific genes, constitutive genes have bigger and stronger NDRs at their transcription start site (TSS) as well as longer distances between both their 5${}^{\prime }$ NDR and TSS, and their 3${}^{\prime }$ NDR and transcription termination site. All these predicted features have been confirmed experimentally with MNase experiments. These two types of genes have different nucleosomal organization resulting in different transcriptional plasticity. In maize, it was also observed that the sequence of constitutive genes has a lower GC content than the sequence of tissue-specific genes, both in introns and exons where it is mainly driven by different codon usage. This likely illustrates selective pressures acting on the nucleosome positioning. The redundancy of the genetic code, allowing the multiplexing of genetic and structural informations [87], is used in this species to promote AT-rich codons in constitutive genes and GC-rich codons in tissue-specific genes, to reduce the GC content of the former and raise the GC content of the latter. This leads to differences in maize genes nucleosomal organization, with a reduced occupancy on constitutive genes, associated with lower transcriptional plasticity. In contrast, the nucleosome occupancy is higher in tissue-specific genes, and associated with higher transcriptional plasticity. This interplay between nucleosome and transcriptional plasticity has also been observed in several other species such as *C. elegans* and *S. cerevisiae*. In *C. elegans*, a time-course of MNase digestion showed that the AT content in the promoter influences nucleosome stability [88]. In this type of experiments, various levels of chromatin digestion are obtained using different concentrations of MNase or different digestion times, providing information about the stability of nucleosomes [52,88,89]. Fragile nucleosomes are identified as nucleosomes only apparent in low-digestion data, as they are more easily destabilized by the MNase than stable nucleosomes [52,88]. Such experiment in *C. elegans* showed that fragile nucleosomes are associated with high AT content of the underlying DNA sequence, and low expression plus high transcriptional plasticity when they are localized at the promoter of genes [52]. In *S. cerevisiae*, it has been shown that genes can be classified according to their nucleosomal organization [55,80,90]. Some genes have a “cristal” nucleosomal organization, with *n* nucleosomes on the body of the genes and a precise, constant NRL. Others have a “bistable” nucleosomal organization, with the possibility to put *n* or n+1 nucleosomes on the body of the gene, the n+1 organization being associated with a higher expression level. These two classes of nucleosomal organization are, like in maize, associated with different transcription plasticity. Indeed, growth genes are associated with “cristal” organization, where stress genes exhibit a “bistable” organization [55,80,90]. Finally, in human, about 70% of promoters are associated to CpG islands (GC rich regions with a CpG dinucleotide content higher than elsewhere on the genome) [91]. These CpG islands have been described to be accessible without the need for ATP-dependent remodeling [92]. This could be due to their DNA sequence inhibiting nucleosome formation, although the well described nucleosome-free region surrounding the TSS of eukarotic genes could also be implicated [93]. All the examples mentioned here show that in a range of organisms, sequence-encoded nucleosomal organization at genes is strongly linked to expression pattern.

Selection of nucleosomal positioning at genes has also been linked to the complexity of organisms (Figure 2) [29,90]. In yeast, the majority of promoter exhibit a NDR, both in vivo and in vitro, indicating that this nucleosomal conformation is encoded directly in the DNA sequence. In contrast, if NDRs can be found in human at the promoter of expressed genes in vivo, it has a rare occurrence in vitro, and sequence-encoded NDRs are typically absent from promoters. In fact, in human, prediction of nucleosomal positioning from sequence showed that promoters are generally occupied by nucleosome attracting regions (NAR), that are the opposite of NDR. One explanation of this difference could lie in the fact that yeasts are unicellular organisms when humans are complex multicellular ones. Most of yeast genes are supposed to be used almost constantly, unlike human genes that are mostly tissue-specific. Following this hypothesis, it could be advantageous for yeast to have a default organization of “open and ready to transcribe” chromatin at their promoter, and to actively close the promoters of the genes that need to be expressed in specific conditions only. In contrast, it could be advantageous in human to adopt the opposite default organization of “closed and repressed” chromatin at promoters and to open specifically the few genes needed in each cell. The comparison of sequence-predicted chromatin conformation at promoters of several species confirmed this hypothesis [29,90]. The nucleosomal organization at promoters follows a gradient, from “mostly NDR” to “mostly NAR”, that corresponds to the complexity of the organisms (identified as the number of different tissues composing the organism) [29]. In other words, yeast, a simple unicellular organisms, exhibited the most sequence-encoded open chromatin at their promoters. Interestingly, the same rule applies in archaea possessing nucleosome-like structures, where the histone core is tetrameric instead of octameric in eukaryotes, leading to the wrapping of only about 80 bp of DNA in archeal nucleosomes instead of 147 bp in eukaryotic nucleosomes [94]. Inversely, vertebrates like zebrafish and mammals, which are multicellular complex organisms, exhibited the most sequence-encoded closed chromatin at their promoters. Between them a range of intermediate signals was found, but with a clear progression from full NDR model for unicellular, to hybrid NDR-NAR and full NAR model in multicellular organisms, according to the increase in organism complexity. This result seems to confirm the hypothesis mentioned earlier about the two models of chromatin at promoter. However, following this hypothesis, genes that are expressed in all cell types of complex multicellular organisms should exhibit a NDR at their promoter, because the “open and ready to transcribe” model would then be advantageous for these genes. Interestingly, this is not the case, and the promoters of these gene are even stronger NAR than cell-type specific genes. To explain this result, it has been proposed that the presence of NAR at promoters could also be linked to a retention of nucleosomes at promoters in cells generally depleted in nucleosomes such as sperm cells, to ensure transmission of epigenetic informations [29]. Regardless of the real biological meaning of these different sequence-encoded nucleosome organizations at promoters, this example shows that it has been modified during the evolution, and that these changes are mainly the result of sequence modifications, with NDR in yeast and NAR in mammals.

### 3.3. Is Chromatin Organization Selected Genome-Wide?

Examples of selection on specific nucleosomal organization at genes through selection of DNA sequences were described in Section 3.2. As nucleosome organization has a direct impact on the expression of genes, either driving expression level or transcriptional plasticity, it has a direct consequence on the fitness of individuals. Hence, selection of the corresponding sequence motifs in the course of evolution makes sense. However, genes represent only a very small fraction of the genome of most multicellular organisms. At numerous loci, nucleosomes are positioned by the intrinsic properties of the DNA sequence on which they are formed (Section 2). For example, nucleosomes are encoded in the DNA sequence over about 37% of the human genome through the statistical positioning at the border of NIEBs [28]. This genome-wide encoding of nucleosomes through nucleosomal barriers seems universal among vertebrates, as predicted in human but also in mouse, cow, pig, chicken and zebrafish [95]. This raises the question of the selection of this nucleosome positioning mechanism. In other words, are nucleosome positions also selected at the genome-wide level? One NIEB feature that is common across vertebrates is the oscillating GC-content profile at NIEB borders, with very low GC at the internal border of NIEBs, then high GC on the ∼140 bp adjacent to the barrier (corresponding to the first stacked nucleosome position), then again low GC over ∼10 bp (first linker), then high GC over the second nucleosome location, low GC on the second linker, and so on. The oscillating pattern becomes less and less pronounced as we move away from the NIEBs, with barely no oscillation detectable after the third nucleosome. However, in the vicinity of NIEBs (∼500 bp of each border), the oscillations are very clear and observed across vertebrates species. As low GC is associated with inhibition of nucleosome formation, and higher GC content in general is associated with nucleosome positioning, the nucleosome organization at the border of NIEBs should also conserved be across these species, through the conservation of GC content. It was indeed observed that there is a link between a higher GC content at the location of nucleosome dyads compared to linker regions and sequence evolution [27,28]. By comparing the interspecies mutations between human and chimpanzee to intraspecies mutations obtained from the 1000 Genomes project [96] in human, several types of selection reinforcing the oscillation of GC content at the border of NIEBs have been observed [28]. First, signature of positive selection for mutations towards A and T nucleotides were described at the internal border of NIEBs and at the linker loci. Inversely, signatures of purifying selection (counterselection) were observed against these mutations at the positions corresponding to nucleosomal DNA. This confirmed an earlier observation of C-to-T mutations favored in linkers and disfavored in nucleosomes [27]. Second, mutations towards G and C nucleotides followed the exact opposite pattern, with purifying selection in NIEBs and in linkers, and positive selection in nucleosomal DNA. Finally, mutations disrupting TTT or AAA sequences (tTt-to-tAt or aAa-to-aTa mutations) were highly counter-selected in NIEBs and linkers, and favored in nucleosomal DNA. As these sequences strongly impair nucleosome formation, this suggests that natural selection is acting on NIEBs to maintain the nucleosomal organization at their borders. In a nutshell, evolution at human NIEBs loci favored mutations towards A and T in non-nucleosomal DNA, and mutations toward C and G in nucleosomal DNA, leading to the oscillating GC content also observed in each vertebrate analyzed, and reinforcing the positioning of two to three nucleosomes at these loci.

### 3.4. Are Transposable Elements Involved in Chromatin Organization?

For now, most studies about the interplay between sequence evolution and nucleosome positioning focused on single nucleotide variations (SNVs), analyzing their position relative to the nucleosomes. However, little is known about other types of mutations such as insertions or deletions in this context. The insertions of transposable elements (TEs) could in fact be important to fully capture the coupling between sequence-mediated nucleosome organization and genome evolution. Indeed, TEs are able to integrate and spread within genomes through a mechanism called transposition [97,98]. They are major components of Eukaryotic genomes, representing for example at least 45% of the human genome [99], although there is a high diversity in terms of TE composition in vertebrate genomes [100]. There are many families of TEs, according to their transposition mechanism, size, DNA base composition, etc. [101]. Some of these elements have been associated to a biological function. For example, a TE insertion can be at the origin of the formation of a new gene, an event called TE domestication (reviewed in [102]). Some TE copies were found to be implicated in various biological processes, for example in the sexual development and function in various animal species [103]. In contrast, some TE insertions have been found to have deleterious effects, with TEs being associated with various diseases [104]. Thus, TEs are major components of the evolution of genomic sequences, their transposition bringing DNA fragments to new locations, inserting from a few tens to several thousands of base pairs of DNA at the insertion site. If the sequence effects of these insertions have been largely investigated such as TFBS transport or coding sequence disruption, the effect of the insertion on the nucleosomal organization remains largely unknown. The insertion of TEs, by disrupting the sequence at the insertion site, could either disrupt or reinforce the sequence-encoded nucleosomal organization, according to the nucleosome-associated properties of the inserted sequence. Thus, apart from being drivers of sequence evolution, TEs could also be drivers of the evolution of nucleosomal organization. Some results already point into this direction such as the presence in human of Alu transposable elements at the border of about half of the NIEBs mentioned in Section 2.3 [95]. The family of Alu TEs is specific to primate genomes [105]. They are short retrotransposons of about 300 bp, with a DNA sequence compatible with the positioning of two nucleosomes [106]. One hypothesis to explain the distribution of Alu TE at the border of human NIEBs is that NIEBs being NDRs and thus accessible to external factors, they could represent preferential target sites for the insertion of Alu TEs. Another hypothesis is that Alu TEs could be at the origin of new NIEBs formation, i.e., nucleosome organization would be a consequence of Alu insertion. Note that these hypothesis are not mutually exclusive, and the link that was observed between NIEBs and Alu TEs in human could result from the interplay between several mechanisms and selection. Moreover, strongly positioned nucleosomes were observed on newly inserted TEs, possibly participating to their regulation [107]. The presence of these nucleosome could both decrease the accessibility to these TEs for transposition machinery, making new transpositions more difficult, and increase the mutation rates on them, because DNA repair is less efficient in nucleosomes than in naked DNA [107]. In a general fashion, there seems to be an interconnection between TEs and the evolution of nucleosomal positioning that still needs to be investigated to fully understand the coupling between sequence evolution and chromatin evolution.

## 4. Feedback of Nucleosomal Positioning on Mutational Patterns

As we saw in Section 3.3, signatures of selection have been clearly identified at the borders of NIEBs. However, another phenomenon could participate to the observed reinforcement of the local GC content. In fact, profiles of mutational rates at NIEB borders were calculated, for both inter- and intra-specific human mutations [28]. This showed for example that interspecies mutation rates towards A and T were higher in non-nucleosomal DNA than in nucleosomal DNA. As discussed above, positive selection would favor these mutations in non-nucleosomal DNA while counterselection would act in nucleosomal DNA. In addition, some oscillations of mutation rates were also observed for intraspecies mutations, for which selection had way less time to influence the mutational pattern. Thus, it seems that even in the presence of weak to no selection, the mutations are not randomly distributed at the borders of the NIEBs. This suggests that nucleosome occupancy has a direct influence on the mutational patterns. The presence of a well-positioned nucleosome, meaning that it almost always covers the same DNA fragment, could then create a mutational bias on this DNA fragment, favoring some mutations type in the nucleosomal DNA with respect to the linker DNA. Nucleosome could bias mutations towards some specific nucleotides on the nucleosomal DNA, by its interaction with DNA damage mechanisms, or the DNA repair machinery. Next generation sequencing progress now permits to establish cartographies of specific DNA damage mechanisms on the genome, and to quantify the efficiency of DNA repair machinery. This made it possible to explore the direct influence of nucleosomes on mutational processes.

Early in the 2000s, it was shown that the excision repair mechanisms of DNA such as base excision repair (BER) or nucleotide excision repair (NER) are hampered by the presence of nucleosomes [108]. It was confirmed a decade later that DNA damages are more persistent in nucleosomal DNA [109]. As DNA damages can lead to mutations, notably during replication, the inhibition of BER and NER has a direct influence on mutational patterns. Nucleosomes also directly modulate the formation rate of certain type of DNA lesions [110]. These properties can be related to the stability of the DNA double helix in the nucleosomal context, as illustrated by the lower degradation rate after cell death of nucleosomal DNA compared to linker DNA in ancient DNA samples [111,112]. Nonetheless, it is crucial to decipher the interplay between nucleosomes and DNA lesion formation and repair mechanisms to understand the influence of nucleosomes on the mutational pattern, and take it into account in evolutionary approaches. Two types of mutational biases have been described in relation to nucleosome positioning [31], associated to nucleosomal occupancy and the rotational positioning of nucleosomal DNA in regards to the histone core, respectively.

Concerning nucleosome occupancy, it has been shown that C-to-T mutations were depleted in nucleosomal DNA relative to linker DNA [113]. As discussed in Section 3.3, natural selection is implicated in the mutational biases [27,28], but a mutational mechanism itself could also be implicated. Indeed, C-to-T mutations usually results from spontaneous deamination of cytosines and 5mCs [31]. This mechanism is more efficient when there is a local opening, called “breathing”, of the DNA double helix. Such breathing of DNA is inhibited in nucleosomal DNA, due to strong structural constraints imposed in the wrapping of DNA around histones, but remains possible in linker DNA, which is free from these constraints [114]. This wrapping is a hindrance to mutations leading to a depletion of the main C-to-T mutations in nucleosomal DNA as compared to linker DNA [113]. Similarly, experiments to map oxidatively induced DNA damages such as 8-oxoguanine (8-oxoG) in *S. cerevisiae* showed that they are modulated by nucleosome occupancy [115]. However, as 8-oxoG persistence depends on the equilibrium between DNA susceptibility to oxidation damage and efficiency of BER, it is still unclear whether the cause of the modulation by nucleosome occupancy is the influence on damage formation or on the efficiency of the repair mechanism [115]. Both hypotheses are not mutually exclusives. Further studies in yeast BER-deficient strains should provide insights about this question.

The effect of nucleosome occupancy on the mutational patterns has also been investigated in cancers where whole genome sequencing of tumors allows to examine the interplay between nucleosomes and mutational signatures [116,117,118]. These signatures correspond to unique combinations of mutation types, generated by specific mutational processes, in one or several types of cancers [119]. For example, mutational signature 1 found in all cancer types results from spontaneous deamination of 5-methylcytosine, and the type of mutation is mainly C-to-T mutation, with preferences for ACG, CCG, GCG and TCG contexts [119]. Mutations from signatures 17 and 18 are mainly T-to-G and C-to-A mutations, respectively, for which the mutational processes involved are unknown. In breast tumors, these two mutational signatures have been found to be more frequent in nucleosomes than it was expected from the sequence composition of the associated DNA fragments [116]. At transcription factor binding sites (TFBSs) flanked by regularly ordered nucleosomes following the model of statistical positioning by anchors (Section 2.3), melanoma mutations (principally induced by UV light) exhibit a periodic distribution associated to nucleosome positioning with a maximal density at nucleosome dyads, which differs from the expected pattern based on sequence composition [120]. More generally, a pan-cancer analysis revealed that for many cancer mutational processes, there are differences in mutation rates between nucleosomal DNA and linker DNA [121]. It also brought new observations, like tobacco-linked mutations occurring more frequently in linker than in nucleosomal DNA. The inhibition of both BER and NER repair systems is hypothesized to be a major player of UV-induced mutational biases. For tobacco-induced mutational bias, the mutational process (bulky DNA adducts at guanines (BPDE-dG)) is known to be inhibited in nucleosomes, leading to the “linker preference” for this type of mutations. The different examples mentioned here show that nucleosome dyad position (the so-called translational positioning of nucleosomes) has an influence on mutational patterns, through the modulation of the efficiency of either the DNA damage processes, or the repair mechanisms, or both, altogether leading to differences in mutation rates and biases between nucleosomal DNA and linker DNA.

Mutations are also modulated at a higher resolution than the nucleosome-linker dichotomy. Indeed, depending on which of the minor or the major groove of a DNA base pair faces the histones (the so-called rotational positioning of DNA within the nucleosome), mutation rates can be variable and, because DNA histone contact points are specifically with the minor groove, each nucleosome translational positioning imposes a specific rotational positioning. A comparison between different *D. melanogaster* populations and between this species and a closely related species showed that C-to-T substitutions were more frequent where the minor groove of DNA faces the nucleosome (minor-in), than where the minor groove of DNA faces away from histones (minor-out) [122]. As at minor-in loci, the DNA is structurally constrained by chemical groups of histones H3 and H4, A/T (or WW) di-nucleotides could be favored for their higher flexibility and low steric hindrance [4]. The periodic occurrence of C-to-T mutations in nucleosomal DNA has been interpreted as a sign of selection on more favorable DNA fragments for nucleosome. However, an alternative hypothesis is that the interaction ability between DNA and mutagenic agents or repair machinery are different at minor-in and minor-out stretches of DNA, resulting in different mutation rates between these loci. This hypothesis is supported by the demonstrated decreased activity of BER at minor-in loci, resulting in lower repair efficiency of methylated guanines, the corollary of this being a higher mutation rate [123]. Experiments with DNase I showed that the accessibility to DNA could be a reason for the decreased activity of BER [124].

Another example of modulation of mutational processes along nucleosomes is for the UV-induced formation of cyclobutane pyrimidine dimers (CPDs) and (6-4) photoproducts (6,4-Pps) in DNA. Both DNA lesions are formed on TT, TC, CT and CC di-nucleotides. In nucleosomal DNA, a ∼10 bp periodicity has been observed in CPDs formation [125]. In fact, this periodic pattern correlates with the rotational positioning of nucleosomes, with preferential CPD formation at minor-out loci [125,126]. The 10 bp periodic pattern and the correlation have been observed genome-wide in yeast and human thanks to a NGS-based damage mapping method named CPD-seq [123,127,128]. The UV-irradiation of the same naked DNA fragment (without nucleosomes) resulted in an opposite CPD formation pattern, with CPDs occurring at positions corresponding to minor-in loci, probably because of the increase of TT dinucleotides at these regions (Section 2) [127]. This means that the underlying sequence is not the cause of the periodic formation pattern of CPDs in nucleosomal DNA, in fact the sequence would even favor the opposite pattern. The presence of a nucleosome, and the structural constraints associated with its formation, override the sequence preferences of CPDs to promote UV-damage at minor-out regions, where the DNA is more accessible. So, nucleosomes have a strong influence on this DNA damage process.

Distribution patterns favoring the minor-out stretches of DNA such as the CPD distribution described above are also found in some types of cancer. In melanoma, the vast majority of somatic mutations are C-to-T transitions at dimers of pyrimidine, characteristic of the UV mutational signature [129]. Analysis of melanoma mutations showed the same ∼10 bp periodicity in well-positioned nucleosomes as the one described for CPD mutations above [121,128]. The same pattern has been retrieved in lung cancer mutations, with high density at minor-out and low density at minor-in [121]. A high resolution genome-wide mapping of DNA damage process implicated in lung cancer mutations would help to understand if it is inhibited at minor-in stretches like CPD formation, or if the DNA damage distribution is constant, which would suggest an implication of the DNA repair mechanisms. On the other hand, other cancers exhibit an opposite mutational pattern, with high mutation densities at minor-in stretches and low mutation densities at minor-out stretches [121]. It has been proposed that a lower efficiency of BER mechanism at minor-in stretches could explain this periodicity. DNA damage at these loci would then be more persistent than at minor-out loci, leading to an increase in mutation rate [121]. Moreover, the presence of nucleosomes impair the recognition of single-strand breaks localized on the non-template DNA strand (NT-SSBs) by poly(adenosine diphosphate-ribose) polymerase 1 (PARP1), and in turn their reparation through BER [130]. Undetected nucleosomal NT-SSBs can be repaired during transcription through transcription-coupled nucleotide excision repair (TC-NER). The efficiency of this reparation mechanism is higher when the NT-SSBs is localized on the side of nucleosomal DNA facing the histone octamer than when it is localized at more accessible loci in nucleosomal DNA. This could be because these accessible loci are more prone to be repaired by BER [130]. Although in the case of NT-SSBs, the interactions between nucleosome and DNA damage and repair mechanisms do not seem to lead to modulated mutation rates, it is another good example of the importance of nucleosome in DNA damage and repair processes.

Hence, the nucleosome has a strong influence on mutational patterns (Figure 3). It affects the effectiveness of excision repair mechanisms like BER and NER [131,132,133,134]. A lot of DNA damages are repaired through BER or NER, like UV-induced, tobacco-induced and oxydative DNA damage. The resulting mutational densities are generally increased in nucleosomal DNA compared to naked linker DNA. However, the activity of BER and NER are also modulated within the nucleosome, with a higher efficiency at minor-out loci, where DNA is more accessible than at minor-in loci, where structural constraints are stronger. The accessibility to DNA inside the nucleosome has also an influence on DNA damage mechanisms, some of them having a preference for minor-out stretches of DNA, and other for minor-in stretches [121]. All these possible influences of nucleosome on mutational patterns need to be taken into account in further attempts to decipher the evolution of DNA sequence regarding nucleosome positioning, to avoid considering as the sign of selection pressure some mutation biases induced by the presence of a nucleosome and its interplay with mutational processes.

## 5. Concluding Remarks

Nucleosome positions in genomes are at least partially encoded in the DNA sequence, through two main mechanisms (Section 2; Figure 1). The first one consists of an interplay between anti-positioning sequences (such as homopolymers like poly(dA:dT)) and high density of nucleosomes, leading to positioning by confinement between nucleosomal barriers [28,55,57,75]. The second mechanism consists in a fine-tuning of nucleosome positioning at the base-pair resolution, with preferences for A/T rich sequences where the DNA is making contact with histone proteins at minor-in positions, and G/C rich sequences where the DNA minor groove is facing away from histones [67]. In vivo and in vitro maps of nucleosomes present high similarities, indicating that the sequence properties are relevant even in the presence of other factors influencing the position of nucleosomes such as ATP-dependent remodelers. As nucleosomes have a functional importance, notably by modulating the accessibility to regulatory regions of genes, sequence evolution should be constrained by the effect of mutations on nucleosome positions (Figure 3). We reviewed several cases of sequence selection for their nucleosomal properties, in yeast, but also in multicellular organisms such as maize and human (Section 3). However, some caveats need to be taken into account when we try to decipher the evolution of sequence regarding its effect on nucleosomes. The first bias that must be considered is the feedback of nucleosome positioning on the mutational patterns. Nucleosomes influence both the mechanisms of DNA damage and DNA repair, leading to difference in the mutational patterns, either between nucleosomal DNA and linker DNA, and within the nucleosomes, between the minor-in and minor-out positions (Section 4; Figure 3). Now that these biases are described, one must be careful interpreting sequence changes in regards to nucleosomal positioning, and properly separate the contribution of selective pressure from the mutational biases induced by the presence of nucleosomes. A second caveat needs to be considered about the interpretation of some observations such as the signature of selective pressure on nucleosomal positioning, as initially raised in [30]. In many studies, mutational data obtained through the comparison of phylogenetically related species assume that the nucleosome organization was identical in the ancestor of the extant species. This “static” view of nucleosome positions in the course of evolution may not be a correct assumption [30], because nucleosomes are frequently repositioned following the evolution of sequences (Section 3). For example, in Eukaryotes, it was observed that A/T-to-G/C mutations are more frequent at the nucleosome dyad. It was interpreted as either a mutational bias caused by the nucleosome, or selection acting on these mutations to reinforce nucleosome positioning, assuming an evolutionary stable nucleosome organization. However, another scenario is compatible with the observations, where the A/T-to-G/C mutations would have repositioned the nucleosomes because of the preference of the dyad for GC-rich motifs [30], i.e., nucleosome positioning would follow the mutations towards G/C (Figure 3). To properly address this possibility, one needs to reconstruct the in vivo nucleosome organization at the time of the mutation. In regions where sequence-encoded nucleosome positioning is relevant in vivo, this can directly be done by applying the nucleosome position prediction tools available (Section 2.4) on the phylogenetically reconstructed ancestral sequences. Otherwise, one would need to compare experimental maps of nucleosome positioning in germline cells of all the species considered. However, for now, just about a handful of species have their nucleosomes mapped experimentally, mainly in somatic or cancer cell lines [54]. Thus, regions where the nucleosomal array appears not to be remodeled, such as NIEB loci in vertebrates, are the best candidates to distinguish between selection, repositioning, and the biased mutation events, and to estimate the relative importance of each mechanism to explain the mutational patterns at nucleosomes.

In this article, we reviewed here findings about the interplay between sequence-encoded nucleosome positioning and evolutionary constraints. Yet, the contribution of the collective properties and functions of the nucleosomal array depending on the position of nucleosomes relative to other nucleosomes have not been addressed. In genomes, the formation of nucleosomal arrays with regularly spaced nucleosomes is conserved across Eukaryotic organisms [135]. These arrays are associated with various functions, such as chromatin condensation in higher order structure, but also with long-range contacts between enhancers and promoters, or inhibition of cryptic transcripts or protection of DNA from double-strand breaks [135]. The formation of nucleosomal arrays depends on various external factors, including remodeling, but also on DNA-binding factors creating nucleosomal barriers against which nucleosomes are stacked, following the model described in Section 2.3. Some sequence motifs such as NIEBs can also act as barriers. If one NIEB does not seem to be sufficient to position more than two to three nucleosomes at each of its borders, two close NIEBs can lead to a regularly spaced array of up to six nucleosomes between them [28]. In vertebrates, the relative position of NIEBs is constrained, with consecutive NIEBs being spaced by distances that are multiple of ∼153 bp [28,95], which was interpreted as a constraint that an integer numbers of compact nucleosomes fits between two close NIEBs. In this way, consecutive NIEBs could form regularly spaced nucleosomal array of controlled NRL following the statistical positioning model between two barriers [55,73,75]. The constraint of NIEB positioning regarding other NIEBs could arise to favor the apparition of such arrays, to use their properties on chromatin condensation and long-range contacts as described above. For example, short NRLs would assure that the intrinsic nucleosome arrays are in an open state, permissive to epigenetic regulation, allowing cell type specific regulation [28]. The influence of DNA sequence on nucleosome positioning and its interplay with the evolution of both sequences and nucleosome positions must then be considered not only at the nucleosome scale, but also at the scale of the nucleosomal array, thus taking into account higher order chromatin structures.

## Figures and Tables

**Figure 1 genes-12-00851-f001:**
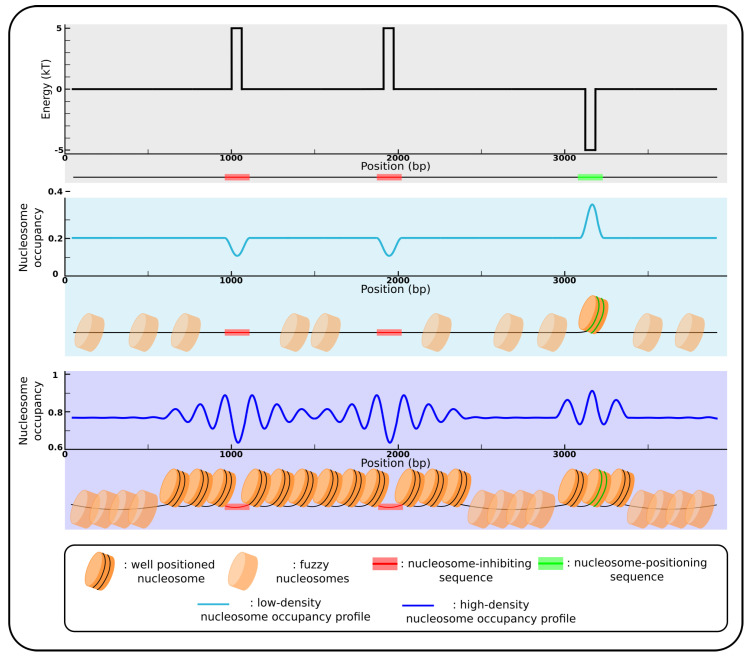
Nucleosomal positioning by sequence motifs. (**Top** panel) Landscape of the energy needed to bend the DNA fragment into the nucleosome depending on the sequence around the nucleosome dyad position (*x*-axis); the hypothetical landscape present two high energy peaks corresponding to two nucleosome-inhibiting sequence motifs (red), and a low energy well at a nucleosome-positioning sequence motif (green). (Mid panel) Nucleosome occupancy profile corresponding to the energy landscape for a low density of nucleosomes. Nucleosomes tend to avoid the inhibiting sequences (as represented by the minima in sky blue curve), and the only preferential nucleosome localisation is at the positioning sequence (peak in sky blue curve). (**Bottom** panel) Nucleosome occupancy profile to corresponding the energy landscape for a high nucleosome density. In this case, nucleosomes still avoid inhibiting sequences (minima in the blue curve), and a global positioning appears between and beside these nucleosomal barriers (oscillations in the blue curve), as a “parking” phenomenon resulting from the non overlapping property of nucleosomes (statistical positioning) (see Section 2.3). Nucleosome positioning also appears beside the well-positioned nucleosome formed on positioning sequence, according to the “anchor-positioning” model described in Section 2.3. Transparent nucleosomes represent fuzzy positioning, meaning that at these loci, nucleosomes have no preferential locations and can be formed more or less anywhere on the DNA.

**Figure 2 genes-12-00851-f002:**
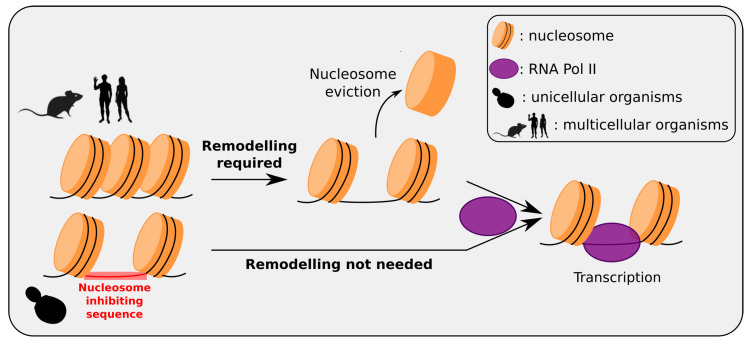
Multicellular/unicellular strategy for intrinsic nucleosomal organization at promoters. In human and most multicellular organisms, typical promoters are intrinsically occupied by nucleosomes, with no inhibiting sequences at these loci; genes are “repressed-by-default” and activated only when needed by the active removal of a nucleosome at the promoter making it accessible to the transcription machinery. In unicellular Eukaryotes such as yeast, inhibiting sequences have been selected at typical gene promoters, to avoid nucleosome formation at these loci; genes are “activated-by-default” since promoters are directly accessible to the transcription machinery, avoiding a remodeling step. These different strategies can be understood as in multicellular organisms, most genes (except housekeeping genes) present tissue-specificity, whereas in a unicellular Eukaryotes most genes are susceptible to be used in every cell. (RNA Pol II is not drawn to scale).

**Figure 3 genes-12-00851-f003:**
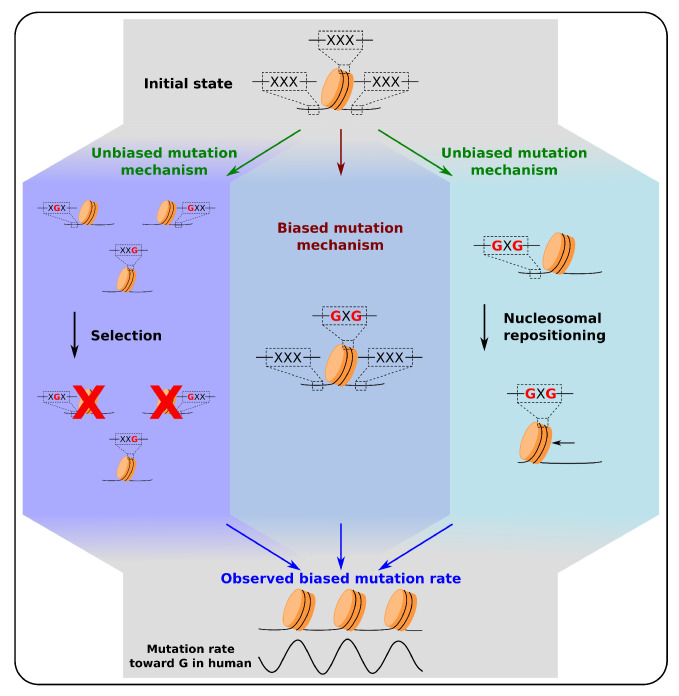
Source of biased mutation rates relative to nucleosomal positioning. Three mechanisms that can lead to biased mutation rates relative to nucleosome positioning as described in Section 3 and Section 4. (**Left**) Mutations that facilitate the positioning of nucleosomes at specific loci are positively selected, those favoring alternative positions are purified. Such mechanism is for example observed at yeast promoters (Section 3.2). (**Center**) A biased mutation mechanism where the presence of a nucleosome drives mutations notably through interactions between nucleosomes and DNA damage and repair mechanisms (Section 4). (**Right**) A nucleosome repositioning model, in which mutations lead to the repositioning of nucleosomes that can also explain the observed biased mutation rates relative to nucleosome positioning when the latter is assumed to remain unchanged during evolution (Section 5). Since all three mechanisms have been observed at the genome scale, the global biased rate of mutations observed is likely to come from a combination of all three mechanisms. The cartoons illustrate possible evolutionary scenarios for 3 trinucleotides (XXX) located in the nucleosomal DNA, and the linker DNA upstream and downstream of a nucleosome. The figure only represents mutations toward G, but these three models are also valid for the other mutational biases (Section 3, Section 4 and Section 5).

## Data Availability

Not applicable.

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
