# Peer review of "Coupling between Sequence-Mediated Nucleosome Organization and Genome Evolution"

_genes, 2021, doi:10.3390/genes12060851_

Round 1
Reviewer 1 Report
In this review Barbier et al. describe the state of the art of the field of nucleosome positioning sequences. The article is well written, important for the field, interesting and easy to read. Authors successfully summarized available information on the topic, making this review a useful tool for other researches.
Comments:
- In the line 34 the word “slightly” in not appropriate. For example, Macro H2A has only 64% homology with H2A, H2A-Bbd - 42% homology.
- In the line 146 “the affinity of the DNA sequence for nucleosome” should be changed to the “the affinity of the DNA sequence for histone core”
- In the line 192 “transcriptional machinery” should be changed to the “ATP-dependent chromatin remodelers”, or it requires further clarification. Otherwise, this statement could lead to misleading interpretation, that RNA Polymerase II changes nucleosome position, which is not the case. It has been shown that it travels through the nucleosome without changing its position, moreover there are mechanisms helping to keep nucleosome order intact after passage of the PolII.
- In the lines 410-411 the statement about archaea needs a citation. The review could benefit, if authors make a bit more detailed comment about archaea chromatin.
- Paragraph 591-606. Distribution of mutations with high density at minor-out and low density at minor-in in some cases could be explained by PolII&nucleosome - dependent mechanism of the DNA breaks recognition, described by Pestov, Gerasimova, et al. Sci Adv. 2015. The review will benefit if this mechanism is mentioned.
- In the paragraph 671-720 information about TE looks more like a part of the main text and not a conclusive remark.
Reviewer 2 Report
This manuscript reports on chromatin structure focusing on the relationship between sequence evolution and nucleosome positioning. Such reviews are needed given the essential role of nucleosomes in biologically significant processes such as transcription, replication, nuclear topology and DNA repair. The text is divided into three large specialized sections plus the Introduction and Conclusions. The ms is well-written, containing sufficient amount of references to the literature with no or little stylistics and grammatical errors. There are several issues that should be considered prior its publication.
General comment
The ms would benefit from reduction of system diversity. I understand that nucleosomes are common features of nuclei all eukaryotic organisms (i.e. fungi, animals and plants) and that it is the authors’ intention to cover whole tree of life. However, information is often redundantly provided for different biological systems. I feel that reduction of examples/biological systems would increase clarity and readability.
Specific comments
Abstract
- The sentence “The DNA sequence is a major factor influencing the position of nucleosomes on genomes“. This is highly debatable since virtually any DNA sequence can be packed into nucleosomes. Positioning signals are rather weak and most chromatins contain nucleosomes at rather relaxed positions. This certainly does not exclude that certain sequence motifs, tend to stimulate nucleosome binding while the others do not or even repulse it. For example, DNA sequences, such as periodic AA and TT dinucleotides, promote nucleosome formation, whereas poly (dA:dT) tracks promote nucleosome-free region. The most generally accepted view is that nucleosome positioning is determined by a combination of DNA sequence, ATP-dependent remodeling enzymes, transcription factors, and elongating RNA polymerases. I feel that these facts should be emphasized and made them more clear in Abstract and throughout the ms.
- Line 44. The sentence “Indeed, actively transcribed genomes where chromatin needs to be open and accessible tend to have shorter NRL….“ While this may be true for some DNA types there are many exceptions from a rule. For example a non transribed telomeric sequence composed of simple motifs (such as TTAGGG) is packed into nucleosomes with extremely short periodicity (<150 bp) in both animals and plants (Mol. Cell. Biol. 14, 5777-5785, 1994; Mol Gen Genet 247, 633-648, 1995; Nucleic Acids Research 48, 5383–5396, 2020)
- Line 368. The sentence : “….that the sequence of constitutive genes has a lower GC content than the sequence of tissue-specific genes. I disagree with this statement since many constitutively expressed genes /also termed housekeeping genes/ contain CpG islands which are generally GC rich (Genes Dev. 25, 1010–1022, 2011) and serve critical functions in promoters and other regulatory regions.
- Positioning is commonly described as translational positioning or rotational positioning (e.g., Gaffney et al., Plos Genet 8(11): e1003036). Translational positioning defines where on a particular stretch of DNA a nucleosome is positioned. Rotational positioning defines the internal orientation of a binding site relative to the nucleosome surface. The difference between these two is fundamental. For example, translational positioning determines if a protein‐binding site is incorporated into a nucleosome and where it is located within the nucleosome.Throughout the text it is often unclear whether the information presented refers to translational or rotation positioning of nucleosomes.
- The Conclusion section is too lengthy and should be reorganized. For example, a whole paragraph about transposable elements starting from line 671 should be cut or markedly reduced since TEs were not discussed in any of the sections. This would leave a space for the discussion of authors’ original ideas presented in Figure 3 and the narrative.
Minor issues
Line 33. “aka H3K9me3“. What is “aka”?
